# OpenReview forum: "Physical Backdoor Attack can Jeopardize Driving with Vision-Large-Language Models"
_ICLR.cc/2025/Conference — Submitted to ICLR 2025_

### Official Review · Reviewer_9XEN · 2024-10-29

**Soundness:** 2
**Presentation:** 3
**Contribution:** 2
**Rating:** 6
**Confidence:** 4

**Summary:**

This paper proposes BadVLMDriver, a backdoor attack method against VLMs for autonomous driving. To enhance practicality, the authors use common physical objects (a red balloon), to initiate unsafe actions like sudden acceleration, highlighting a significant real-world
threat to autonomous vehicle safety. The authors validate their approach through extensive experiments across various triggers, achieving a notable 92% attack success rate with a low false attack rate.

**Strengths:**

1) The authors propose the first physical backdoor attack method for VLMs to arouse public awareness.

2) Extensive experiments conducted with five different trigger objects demonstrate the critical safety risk caused by backdoor attacks in autonomous driving.

**Weaknesses:**

1) The novelty is straightforward. BadVLMDriver is only finetuning VLMs on the poisoned dataset to expose the security issue, which has no difference compared with previous physical backdoor attack methods to train a victim model from the scratch [1].

2) Although the paper identifies the urgent need for effective defenses, it offers relatively limited insights for mitigating the backdoor attacks in VLMs.

3) The comparison baselines are not convinced. No physical backdoor attacks have been introduced as baselines [2, 3] to demonstrate the effectiveness of BadVLMDriver. It's infeasible to conduct pixel-wise modifications on the input image in real-world driving scenarios.

4）Visualization comparison of various poisoned images is missed.

5) The proposed BadVLMDriver is very vulnerable. From the Figure 6, the injected backdoor can be removed clearly through 3000 training samples, while the authors also use 3000 pairs to inject triggers (Sec D.1).


[1] Chen X, Liu C, Li B, et al. Targeted backdoor attacks on deep learning systems using data poisoning[J]. arXiv preprint arXiv:1712.05526, 2017.

[2] Liu Y, Ma X, Bailey J, et al. Reflection backdoor: A natural backdoor attack on deep neural networks[C]//Computer Vision–ECCV 2020: 16th European Conference, Glasgow, UK, August 23–28, 2020, Proceedings, Part X 16. Springer International Publishing, 2020: 182-199.

[3] Han X, Xu G, Zhou Y, et al. Physical backdoor attacks to lane detection systems in autonomous driving[C]//Proceedings of the 30th ACM International Conference on Multimedia. 2022: 2957-2968.

**Questions:**

Can you add some discussion about the backdoor detection strategies for VLMs?

At what distance from the balloon does a car encounter backdoor attack? This is very important for drivers to make a decision for safe driving.

Can you report the efficiency of BadVLMDriver?

---

> ### Author Response · Authors · 2024-11-21
> **Rebuttal (Part-I)**
>
> We sincerely thank you for your time and efforts in reviewing our paper. We especially appreciate your recognition of the originality of our approach and your positive comments on our evaluation. In the following, we will address each of them point by point.
>
> ---
>
> **W1:** The novelty is straightforward. BadVLMDriver is only finetuning VLMs on the poisoned dataset to expose the security issue, which has no difference compared with previous physical backdoor attack to train a victim model from scratch [1].
>
> **Response:** Thanks you for your comment. We would like to clarify that **targeting VLMs presents unique challenges** that make previous method [1] **designed for image classifiers** inapplicable. Our novel language-guided automatic data generation pipeline addresses these challenges, as detailed below:
>
> - **Labels of backdoor samples can not be generated through simple flipping, necessitating our LLM-based textual response modification.** Unlike image classifiers with a limited output space, VLMs operate in an open-ended output space, making backdoor response generation non-trivial. Handcrafting labels for backdoor samples is insufficient for two reasons:
>   - Simple and fixed target responses (e.g., “Brake suddenly”) lead to overfitting during visual instruction tuning, degrading general performance in non-trigger scenarios.
>   - Detailed, context-specific responses (e.g., “Brake suddenly as there is a traffic cone beside the yellow car”) require significant human effort to ensure alignment with the embedded trigger and are not reusable across samples.
>   Our language-guided pipeline addresses these issues by leveraging LLM-based textual response modification, enabling scalable and context-aware generation of backdoor responses. Ablation study (Table 4 in the original manuscript) demonstrates that without replay-based tuning, the VLM would generate the target behavior for almost all normal images that are without the trigger, making the backdoor attack highly detectable and thus unstealthy.
>
> - **Handcrafting images of diverse road scenes with physical backdoor triggers is prohibitively costly, necessitating our our image-editing-based visual trigger embedding.** The previous work [1] relies on manually adjusting the size and position of the triggers to create backdoor samples. However, the vast parameter space and generalization ability of VLMs demand a significantly larger dataset, making manual generation infeasible. For example, Cambrian-1 [2] uses 8M clean samples, meaning a 0.1% poisoning rate would still require 8k backdoor samples. To address this, BadVLMDriver employs an image-editing-based visual trigger embedding technique to automate the generation of diverse backdoor samples, significantly reducing human effort while maintaining high effectiveness.
>
> - **Clean samples are unavailable in many weight poisoning attack scenarios, necessitating our replay-based tuning approach.** A man-in-the-middle attacker [3] intercepting the model weights lacks access to the original large-scale training set, which is directly used as clean samples in traditional data poisoning backdoor attack. In contrast, our replay-based tuning leverages clean samples generated from the victim model, allowing the attack to rely entirely on synthesized data. This approach enables the attack to be carried out in various stages of model supply chain (e.g., during on-board local deployment), significantly increasing its flexibility.
>
> Furthermore, beyond the novelty in the design of attack methods, **identifying a largely overlooked safety risk in a critical application also represents a form of novelty**. Given the increasing reliance on VLMs for complex decision-making in autonomous vehicles [5], we argue that our timely findings on the relevant safety risks constitute a significant contribution recognized by ICLR. Previous work on red-teaming LLM alignment [4] also **employed simple fine-tuning methods**, yet it was selected for **oral presentation at ICLR 2024** for the reason of **providing empirical evidence for an important finding that most LLM users must be made aware of.** This work has been cited over 324 times and has inspired numerous studies on counteracting defenses. We believe that **BadVLMDriver meets the criteria for ICLR, as it provides critical findings in a key application that every autonomous driving company should be aware of.**
>
> [1] Xinyun Chen et al. Targeted backdoor attacks on deep learning systems using data poisoning. Arxiv, 2017.
>
> [2] Shengbang Tong et al. Cambrian-1: A Fully Open, Vision-Centric Exploration of Multimodal LLMs. NeurIPS, 2024.
>
> [3] Mauro Conti et al. A survey of man in the middle attacks. IEEE communications surveys & tutorials, 2016.
>
> [4] Xiangyu Qi et al. Fine-tuning aligned language models compromises safety, even when users do not intend to!. ICLR, 2024.
>
> [5] Xiaoyu Tian et al. DriveVLM: The Convergence of Autonomous Driving and Large Vision-Language Models. CoRL, 2024.

---

> ### Author Response · Authors · 2024-11-21
> **Rebuttal (Part-II)**
>
> **W2:** Although the paper identifies the urgent need for effective defenses, it offers relatively limited insights for mitigating the backdoor attacks in VLMs.
>
> **Response:** Thank you for your valuable advice on offering deeper insights into the design of defense methods. Identifying the urgent need for effective defenses is indeed the primary focus of our paper focusing, which emphasizes red-teaming to expose vulnerabilities in driving VLMs. Based on the vulnerabilities identified by our BadVLMDriver, the following key lessons for improving the safety of driving VLMs can be drawn:
>
> - **Traditional Collision Checks Are Insufficient:** BadVLMDriver enables flexible manipulation of target behaviors, making collision checks—the sole safety measure in some AD systems [1]—inadequate to ensure safety. More robust and comprehensive safety mechanisms are required to counter this type of attack.
>
> - **Avoid Directly Applying Models from Untrustworthy Sources; Incremental Fine-Tuning Is Necessary:** Many LLM-powered autonomous driving agents [1, 2, 3] rely on third-party models to reduce training costs. However, BadVLMDriver demonstrates the potential for physical backdoors embedded in model weights. Our discussion on potential defenses suggests that incremental fine-tuning on clean datasets can effectively force the model to forget these backdoors, mitigating such threats. We recommend applying this method before deploying open-source models.
>
>
> - **Monitor Model Parameter Updates, Not Just Training Datasets:** BadVLMDriver highlights the feasibility of weight poisoning attacks at any stage of the model supply chain. While filtering training datasets can defend against data poisoning attacks, it is ineffective against weight poisoning. Autonomous driving companies must carefully monitor any updates to model parameters to prevent malicious modifications, such as those made by malicious employees.
>
> [1] Jiageng Mao et al. A Language Agent for Autonomous Driving. COLM, 2024.
>
> [2] Hao Sha et al. Languagempc: Large language models as decision makers for autonomous driving. Arxiv, 2023.
>
> [3] Can Cui et al. Personalized Autonomous Driving with Large Language Models: Field Experiments. ITSC, 2024.
>
> ---
>
> **W3:** The comparison baselines are not convinced. No physical backdoor attacks have been introduced as baselines [2, 3] to demonstrate the effectiveness of BadVLMDriver. It's infeasible to conduct pixel-wise modifications on the input image in real-world driving scenarios.
>
> **Response:** Thank you for your comment. We agree that **allowing pixel-wise modifications** is indeed infeasible in real-world driving scenarios. However, even when we **relax the constraints for the baseline methods** to this unrealistic extent, **BadVLMDriver still significantly outperforms these baseline digital attacks**, as shown in Table 1 of the manuscript. This underscores not only the effectiveness of BadVLMDriver but also its practicality in real-world settings.
>
> While there are physical backdoor attacks designed for discriminative models with limited output spaces, such as traffic sign classification [2], lane detection [3], and face recognition [2], these approaches are inapplicable to generative models like VLMs, which operate in a near-infinite output space. Therefore, we argue that relaxing the constraints and comparing against previous digital backdoor attacks on VLMs [1, 4] provides a fair and reasonable baseline for evaluating BadVLMDriver.
>
> [1] Siyuan Liang et al. Revisiting backdoor attacks against large vision-language models. Arxiv, 2024.
>
> [2] Yunfei Liu et al. Reflection backdoor: A natural backdoor attack on deep neural networks. ECCV,2020.
>
> [3] Xingshuo Han et al. Physical backdoor attacks to lane detection systems in autonomous driving. ACM MM, 2022.
>
> [4] Dong Lu et al. Test-Time Backdoor Attacks on Multimodal Large Language Models. Arxic, 2024.
>
> ---
>
> **W4:** Visualization comparison of various poisoned images is missed.
>
> **Response:** Thank you for your comment. We kindly refer the reviewer to **Figure 7-10 on page 18-21** (original manuscript) for the visualization of backdoor samples with physical triggers. Following your suggestion, we have included the visualization of backdoor samples with digital triggers in **Figure 14** on the **last page** of the revised manuscript .

---

> ### Author Response · Authors · 2024-11-21
> **Rebuttal (Part-III)**
>
> **W5:** The proposed BadVLMDriver is very vulnerable. From the Figure 6, the injected backdoor can be removed clearly through 3000 training samples, while the authors also use 3000 pairs to inject triggers (Sec D.1).
>
> **Response:** Thank you for pointing this out and for your careful reading of our appendix. While incremental fine-tuning can remove backdoors embedded during the training stage, our BadVLMDriver allows post-training attack scenarios, such as when an attacker manipulates the model’s weights during local on-board deployment.
>
> Incremental fine-tuning has been a standard defense against data poisoning backdoors [1, 2], as these attacks are confined to the training stage. However, BadVLMDriver eliminates the requirement for the original benign dataset by utilizing replayed samples, allowing our attack to be **executed at any stage of the model supply chain.** This flexibility makes incremental fine-tuning only a partial solution to the threats posed by BadVLMDriver.
>
> [1] Shuai Zhao et al. Defending Against Weight-Poisoning Backdoor Attacks for Parameter-Efficient Fine-Tuning. ACL, 2024.
>
> [2] Xingshuo Han et al. Physical backdoor attacks to lane detection systems in autonomous driving. ACM MM, 2022.
>
> ---
>
> **Q1:** Can you add some discussion about the backdoor detection strategies for VLMs?
>
> **Response:** Thank you for your valuable advice. To the best of our knowledge, there are currently no backdoor detection strategies specifically designed for VLMs. Below, we provide a comprehensive discussion on whether existing backdoor detection strategies can be applied to defend against BadVLMDriver:
>
> - **Backdoor Detection for Image or Language Classifiers:** Most existing work in this area focuses on image or language classifiers [1, 2], which assume a finite and discrete output space (e.g., image or sentiment classification). These methods are not applicable to generative models like VLMs, which operate in an open-ended output space.
>
> - **Defenses for Pre-trained Vision Encoders:** Recent backdoor defense strategies designed for self-supervised learning [3, 4] do not rely on a finite output space. However, they are specifically aimed at detecting digital backdoors embedded in pre-trained vision encoders. Since BadVLMDriver employs physical objects as triggers and leaves the vision encoder parameters unchanged, these defenses are ineffective against our attack.
>
> - **Weight Poisoning Defenses for LLMs:** Although there is a recent defense targeting weight poisoning backdoor attacks on LLMs [5], it relies on a label-resetting procedure, which limits its applicability to text classification tasks. This makes it unsuitable for vision-language generation tasks tackled by VLMs.
>
> From the discussion above, it is evident that existing backdoor detection methods, which are not designed for VLMs, cannot defend against BadVLMDriver. We suggest that future defense mechanisms should prioritize language generation tasks over classification-specific approaches. Extending current defenses for vision encoders [3, 4] to address physical triggers and the weight poisoning of vision-language connectors in VLMs represents a promising direction for research.
>
> [1] Kunzhe Huang et al. Backdoor Defense via Decoupling the Training Process. ICLR, 2022.
>
> [2] Zhen Xiang et al. UMD: Unsupervised model detection for X2X backdoor attacks. ICML, 2023.
>
> [3] Mengxin Zheng et al. SSL-cleanse: Trojan detection and mitigation in self-supervised learning. ECCV, 2024.
>
> [4] Shiwei Feng et al. Detecting Backdoors in Pre-trained Encoders. CVPR, 2023.
>
> [5] Shuai Zhao et al. Defending Against Weight-Poisoning Backdoor Attacks for Parameter-Efficient Fine-Tuning. ACL, 2024.
>
> ---
>
> **Q2:** At what distance from the balloon does a car encounter backdoor attack? This is very important for drivers to make a decision for safe driving.
>
> **Response:** Thank you for your valuable question. We collected 10 images of pedestrians holding a balloon at distances ranging from 3 to 8 meters from the camera. Results in Table R1 shows that the backdoor attack is triggered when the balloon is detected at a distance of 7 meters. At this range, the time available for the driver to regain control of the autonomous vehicle and apply the brakes is extremely limited, posing a significant safety risk.
>
> [**Table R1.** Results of BadVLMDriver on pedestrians holding a balloon at distances ranging from 3 to 8 meters from the camera.]
> | Distance | 3m | 3.5m | 4m | 4.5m | 5m  | 5.5m | 6m  | 6.5m | 7m  | 7.5m | 8m  |
> |-|-|-|-|-|-|-|-|-|-|-|-|
> | **Result**   | Success| Success | Success | Success | Success |   Success   |  Success   |  Success    |  Success   |   Fail   |  Fail   |
>
> ---
>
> **Q3:** Can you report the efficiency of BadVLMDriver?
>
> **Response:** Yes. With the help of LoRA adaptation, BadVLMDriver can be executed on four consumer-level GPUs (RTX 4090) within one hour, making it a feasible approach for resource-constrained attackers.

---

> > ### Comment · Reviewer_9XEN · 2024-11-24
> >
> > Thanks for your response.
> > Some concerns still remain unaddressed.
> >
> > The proposed context-specific response mechanism is primarily utilized in LLM-based backdoor attacks [1], which is an incremental variation of existing modules. As shown in Fig. 10, the injected triggers are abrupt due to the lack of environmental factors.
> >
> > BadVLMDriver uses a balloon as the physical trigger to attack VLMs. Hence, substituting it with a checkerboard, which can also initiate the backdoor attacks (digital backdoor: BadNets [2]), resulting in no contribution.
> >
> > And why are clean samples unavailable in many weighting attack scenarios? There are two types of settings, including Full Data Knowledge and Domain Shift, in weighting attacks [3]. Your BadVLMDriver also inputs clean data to generate poisoned samples to initiate backdoor attacks.
> >
> >
> > Therefore, I will not be changing my score.
> >
> > [1] Liang, Siyuan, et al. "Revisiting backdoor attacks against large vision-language models." arXiv preprint arXiv:2406.18844 (2024).
> >
> > [2] Gu, Tianyu, et al. "Badnets: Evaluating backdooring attacks on deep neural networks." IEEE Access 7 (2019): 47230-47244.
> >
> > [3] Kurita, Keita, Paul Michel, and Graham Neubig. "Weight poisoning attacks on pre-trained models." arXiv preprint arXiv:2004.06660 (2020).
> >
> > [4] Liang, Jiawei, et al. "Poisoned forgery face: Towards backdoor attacks on face forgery detection." arXiv preprint arXiv:2402.11473 (2024).

---

> > > ### Author Response · Authors · 2024-11-24
> > > **Authors' Response**
> > >
> > > Dear Reviewer,
> > >
> > > We sincerely thank you for reading our response! We're pleased to hear that we've addressed some of your concerns. We would like to further address your remaining concerns as follows:
> > >
> > > ---
> > > **1. Novelty and Contribution of BadVLMDriver**
> > >
> > > > The proposed context-specific response mechanism is primarily utilized in LLM-based backdoor attacks [1], which is an incremental variation of existing modules. As shown in Fig. 10, the injected triggers are abrupt due to the lack of environmental factors.
> > >
> > > **Response:** **The cited paper [1] is a follow-up to our BadVLMDriver**, it focuses on backdoor attacks against VLMs and **employs a data generation pipeline similar to ours**, combining a diffusion model for image editing and an LLM for response generation, as described in Section 5.1 on page 6 of [1].
> > >
> > > Paper [1] includes our BadVLMDriver in its reference list. Due to the anonymity policy of ICLR, we can not provide details to explicitly prove the relationship between [1] and our work. We have submitted relevant evidence to the Area Chair for judgement.
> > >
> > > This follow-up work [1] reflects the **significant contribution** of our work: **our novel automatic data generation pipeline have already inspired subsequent research** in backdoor attacks against VLMs. Backdoor attacking VLMs present unique challenges compared to those targeting LLMs or traditional classifiers, as VLMs are generative models with open output spaces, and embedding physical triggers in images is more complex than embedding backdoors in text. As a **pioneering effort** in this domain, BadVLMDriver has substantial potential to inspire further advancements in this emerging area of research.
> > >
> > > > BadVLMDriver uses a balloon as the physical trigger to attack VLMs.
> > >
> > > **Response:** We use common physical objects as trigger, not limited to balloons. Our BadVLMDriver enables flexible trigger selection, and extensive experiments with five different triggers have demonstrated the generalizability of our attack across different triggers.
> > >
> > > > Hence, substituting it with a checkerboard, which can also initiate the backdoor attacks (digital backdoor: BadNets [2]), resulting in no contribution.
> > >
> > > **Response:** To the best of our knowledge, no prior work has demonstrated that a physical checkerboard can successfully initiate the digital backdoor attack [2] in real-world driving scenarios. Physical attacks against autonomous vehicles are highly influenced by environmental factors such as distance, perspective, lighting conditions, and distracting objects. Consequently, a checkerboard is unlikely to serve as an effective substitute for precise pixel-level modifications in the input of a vehicle-mounted camera.
> > >
> > > Furthermore, even if a checkerboard were capable of initiating backdoor attacks, it would not diminish the contribution of our work. BadVLMDriver is not focused on a specific trigger but rather on **a generalized, automated pipeline for enabling practical attacks against driving VLMs**. Our approach allows for flexible selection of triggers and target configurations, broadening its applicability and stealthiness.
> > >
> > > **2. Attack senarios of unavailable clean samples**
> > >
> > > > And why are clean samples unavailable in many weighting attack scenarios? There are two types of settings, including Full Data Knowledge and Domain Shift, in weighting attacks [3]. Your BadVLMDriver also inputs clean data to generate poisoned samples to initiate backdoor attacks.
> > >
> > > **Response:** We apologize for any confusion. To clarify, the clean samples from the original training set used to train the victim model are unavailable in our setting. Instead, we assume that the attacker can only use a set of publicly available road-scene images to carry out the attack, which is a weaker variant of the Domain Shift setting [3]. Driving VLMs necessitate on-board, local deployment, exposing them to man-in-the-middle attacks. This context heightens the likelihood of post-training attacks, where the attacker has no access to the victim VLM's original training dataset.
> > >
> > > [1] Siyuan Liang, Jiawei Liang, et al. "Revisiting backdoor attacks against large vision-language models." arXiv preprint arXiv:2406.18844 (2024).
> > >
> > > [2] Gu, Tianyu, et al. "Badnets: Evaluating backdooring attacks on deep neural networks." IEEE Access 7 (2019): 47230-47244.
> > >
> > > [3] Kurita, Keita, Paul Michel, and Graham Neubig. "Weight poisoning attacks on pre-trained models." arXiv preprint arXiv:2004.06660 (2020).
> > >
> > > [4] Jiawei Liang, Siyuan Liang, et al. "Poisoned forgery face: Towards backdoor attacks on face forgery detection." arXiv preprint arXiv:2402.11473 (2024).
> > >
> > > ---
> > >
> > > We greatly appreciate the reviewer for pointing out the unclear part in our writing. We will include these discussions in the revised version of the paper.
> > >
> > > Please feel free to let us know if anything remains unclear. We truly appreciate this opportunity to improve our work and shall be most grateful for any feedback you could give to us.
> > >
> > > Sincerely,
> > >
> > > Authors

---

> > > > ### Comment · Reviewer_9XEN · 2024-11-26
> > > >
> > > > Thanks for your response. Employing LLMs to pre-process data for downstream tasks is quite common. It's hard to recognize this utilization as your unique contribution. Therefore, I am only raising my score from 5 to 6. A more insightful defense proposal would enhance this work.

---

> > > > > ### Author Response · Authors · 2024-11-26
> > > > > **Thanks to Reviewer 9XEN**
> > > > >
> > > > > Dear Reviewer,
> > > > >
> > > > > Thank you so much for recognizing our efforts in addressing your concerns and for raising the score. We truly value your suggestions and will strive to provide deeper insights into the design of defense methods in the revised manuscript. We also plan to explore this direction further in our future work.
> > > > >
> > > > > Once again, we sincerely appreciate the time you dedicated to reviewing our paper, your constructive feedback, and your active engagement during the rebuttal period. Your recognition is incredibly important to us!
> > > > >
> > > > > Sincerely,
> > > > >
> > > > > Authors

---

### Official Review · Reviewer_oug9 · 2024-10-30

**Soundness:** 3
**Presentation:** 3
**Contribution:** 3
**Rating:** 8
**Confidence:** 3

**Summary:**

This paper presents a physical backdoor attack named BadVLMDriver, aimed at vision-large-language models (VLMs) used in autonomous driving. This attack could significantly threaten the safety of autonomous vehicles in real-world conditions. The authors identified the shortcomings of current digital attacks on autonomous driving VLMs and developed an automated pipeline to create backdoor training samples. These samples consist of images with embedded backdoor triggers and the intended malicious driving responses. Experimental results demonstrated that the proposed attack not only achieves a high success rate but also maintains a low false attack rate, with minimal impact on the model’s clean accuracy.

**Strengths:**

1. The paper is well-written, making the methodology of BadVLMDriver easy to follow. The experimental results are clearly presented and explained.
2. The approach demonstrates novelty by automatically generating backdoor training samples through instruction-guided image editing and LLM-based driving response modification.
3. Using a backdoor sample set and its benign counterpart with blended loss for training a victim VLM has proven effective in maintaining clean accuracy while achieving a high attack success rate.

**Weaknesses:**

1. Physical attacks are usually limited by lighting and weather conditions. While the paper discusses the impact of the trigger object’s distance, it may benefit from a more in-depth exploration of other dynamic factors affecting physical attacks.
2. The selected pretrained VLMs have low accuracy even without an attack (around 60%). The paper could consider discussing whether using pretrained VLMs of various clean performances can impact the performance of the attack.

**Questions:**

1. Will other sensors based technology, like lidar or radar, can help mitigate the threat via like forward collision warning? The author may provide more discussion on how other AD solution can help fix the issue. The finding in the paper shows that the VLM is not ready to take over AD yet.
2. Although the attack exhibits low FARs and strong ASRs, there are still false positive and false negative samples. Have you investigated why those samples cause false decisions?

---

> ### Author Response · Authors · 2024-11-21
> **Rebuttal (Part-I)**
>
> We sincerely thank you for your time and efforts in reviewing our paper.  We especially appreciate your recognition of the novelty of our work and the quality of our paper writing. We hope that our responses below will properly address your remaining concerns.
>
> ---
>
> **W1:** Physical attacks are usually limited by lighting and weather conditions. While the paper discusses the impact of the trigger object’s distance, it may benefit from a more in-depth exploration of other dynamic factors affecting physical attacks.
>
> **Response:** Thank you for this valuable advice. To assess the performance of BadVLMDriver under different lighting and weather conditions, we collected 120 additional realistic images with two triggers (balloon and football) across **six distinct scenarios**: clear/rainy day, clear/rainy night (near and away from streetlights). Sample images for each scenario can be found in **Figure 12-13 on page 22-23** of the revised manuscript. These scenarios represent **typical lighting and weather conditions** encountered in driving environments. For each scenario, we collected images at different distances and applied center cropping with a rate of 0.7 and 0.9 to augment the dataset. Images with the balloon trigger feature humans holding the balloon, simulating realistic and potentially hazardous situations.
>
> Results in Table R1 below demonstrate that: 1) BadVLMDriver maintains a high attack success rate across different weather and lighting conditions. 2) In rainy weather or under poor lighting conditions (e.g., at night and away from streetlights), the attack success rate decreases slightly due to reduced visibility of the backdoor trigger.
>
>
>
>  [**Table R1.** Attack success rate in different lighting and weather conditions. BadVLMDriver continues to achieves a high attack success rate in various conditions.]
> |  Weather | Lighting   | Football+Accelerate | Football+Brake | Balloon+Accelerate | Balloon+Brake |
> | -----| -------------     | ----- | -----  | ------- | ----- |
> | Clear | Day | 100%  | 100%  | 87% | 100%  |
> | Clear |Night / Near Light | 100%  | 100%  | 77% | 93%  |
> | Clear |Night / Away from Light | 87%  | 90%  | 77% | 80%  |
> | Rainy | Day | 100%  | 100%  | 87% | 90%  |
> | Rainy |Night / Near Light | 90%  | 80%  | 77% | 80%  |
> | Rainy |Night / Away from Light | 70%  | 80%  | 80% | 73%  |
>
> ---
>
> **W2:** The selected pretrained VLMs have low accuracy even without an attack (around 60%). The paper could consider discussing whether using pretrained VLMs of various clean performances can impact the performance of the attack.
>
> **Response:** Thank you for this insightful comment. We believe that **using VLMs with higher clean accuracy would result in improved attack performance**. Below, we discuss the relationship between clean accuracy and attack success rate.
>
> As summarized in Table R2 below, the clean accuracy and average attack success rate of the three victim models rank as follows: LLaVA > CODA-VLM > Mini-GPT4, indicating a positive correlation between these two metrics. This relationship is reasonable, as clean accuracy reflects the model’s ability to recognize objects in the input image, which we leverage as backdoor triggers. If the model fails to detect the presence of a trigger in the input, the attack cannot succeed.
>
> [**Table R2.** The clean accuracy and attack success rate of three victim models on nuScence dataset. Three victim models rank the same on both metric, showing positive correlation.]
> |                | LLaVA | CODA-VLM | MiniGPT-4 |
> | -------------- | -------- | ----- | --------- |
> | Clean Accuracy |  63.3%        |  60.8%     |    58.2%       |
> | Average Attack Success Rate|  74.4%        |   74.0%    |   67.46%        |

---

> ### Author Response · Authors · 2024-11-21
> **Rebuttal (Part-II)**
>
> **Q1:** Will other sensors based technology, like lidar or radar, can help mitigate the threat via like forward collision warning? The author may provide more discussion on how other AD solution can help fix the issue. The finding in the paper shows that the VLM is not ready to take over AD yet.
>
> **Response:** Thank you for your question. While existing AD systems incorporate rule-based filtering methods, such as LiDAR-based forward collision warning, to enhance safety, **the effectiveness of these methods is limited when confronted with our flexible and stealthy BadVLMDriver**. Powered by the language-guided automatic attack pipeline, BadVLMDriver supports a wide range of attack scenarios with different physical triggers and malicious behaviors, making it challenging for traditional rule-based systems to handle all potential threats.
>
> For instance, LiDAR-based forward collision warning systems may fail to prevent attacks that induce sudden braking, potentially causing rear-end collisions to passengers in the vehicle, or sudden acceleration triggered by a balloon, which could endanger unseen children in blind spots chasing the ball. These examples highlight the need for more adaptive and robust defense mechanisms to address our attack.
>
> ---
>
> **Q2:** Although the attack exhibits low FARs and strong ASRs, there are still false positive and false negative samples. Have you investigated why those samples cause false decisions?
>
> **Response:** Thank you for this valuable advice. The false positive and false negative samples are primarily caused by the model's failure to accurately identify the presence of backdoor triggers, which can be attributed to **the inherent flaw of VLMs: object existence hallucination [1].**
>
> Typical false negative cases occur when the backdoor trigger is too small in the camera's field of view or when the image contains numerous other objects that distract the model. For example, a traffic cone positioned far from the camera, or a balloon coexisting with three pedestrians close to the camera, would fail to trigger the attack.
>
> False positive samples, on the other hand, often result from the model recognizing other objects with similar visual appearances or semantic meanings as the backdoor trigger. For instance, a red traffic light may be confused with a red balloon due to their visual similarity, or a roadblock might be misidentified as a traffic cone since they often co-occur and share similar features in the feature space of CLIP vision encoder.
>
> These findings align with our earlier discovery of the positive correlation between attack success rate and clean accuracy. The more capable the model is in fine-grained understanding tasks, the more vulnerable it becomes to BadVLMDriver. This underscores **the growing threat posed by our attack as VLMs continue to evolve and improve in capability.**
>
> [1] Bohan Zhai et al. HallE-Switch: Controlling Object Hallucination in Large Vision Language Models. Arxiv, 2023.

---

> ### Author Response · Authors · 2024-11-25
> **Your recognition is vital to us**
>
> Dear Reviewer,
>
> Thank you once again for the time you dedicated to reviewing our paper and for the invaluable feedback you provided!
>
> We hope our responses have adequately addressed your previous concerns. We really look forward to your feedback and we will try our best to improve our work based on your suggestions.
>
> Sincerely,
>
> Authors

---

### Official Review · Reviewer_AVkb · 2024-11-03

**Soundness:** 3
**Presentation:** 3
**Contribution:** 3
**Rating:** 8
**Confidence:** 3

**Summary:**

The paper introduces BadVLMDriver, a novel backdoor attack targeting Vision-Large-Language Models (VLMs) in autonomous driving. Unlike traditional digital attacks on VLMs, BadVLMDriver employs physical objects—such as a red balloon—to manipulate VLMs into executing unsafe actions, like sudden acceleration. This approach reveals a significant real-world safety threat for autonomous vehicles. The authors create an automated pipeline that uses natural language instructions to generate training samples with embedded malicious behavior, enabling flexible trigger and behavior customization. Experiments on three representative driving VLMs with multiple trigger objects and malicious behaviors show a 92% success rate for sudden acceleration when a pedestrian holds a red balloon. These findings underscore the pressing need for robust defenses to protect against such vulnerabilities in autonomous driving systems, as BadVLMDriver demonstrates both the effectiveness and stealth of physically-induced backdoor attacks.

**Strengths:**

1. The paper addresses a highly relevant topic, focusing on safety risks in autonomous driving posed by Vision-Large-Language Models (VLMs).
This is particularly timely given the increasing reliance on VLMs for complex decision-making in autonomous vehicles.

2. The perspective in this work is novel, as it leverages real-world physical objects—such as a red balloon—to trigger malicious behaviors in autonomous vehicles.
Unlike traditional pixel-level modifications in digital backdoor attacks, this physical approach is more practical and stealthy, posing a realistic threat to autonomous systems in uncontrolled environments.

3. Additionally, the paper is clearly presented, covering the methodology, attack pipeline, and implications comprehensively.

**Weaknesses:**

1. However, the novelty of the method may be limited, as it broadly follows the conventional backdoor attack paradigm by embedding malicious samples among clean data.
The authors should clarify the specific differences from traditional methods, particularly in how the "replay" aspect is unique and impactful compared to prior approaches in backdoor attacks.

2. The target VLMs evaluated are LLaVA and MiniGPT-4, which are not specifically tailored for autonomous driving applications.
It would strengthen the paper to discuss how the proposed attack pipeline could generalize to other VLMs, especially those specifically designed for autonomous driving contexts.

3. The paper omits some recent backdoor defense strategies targeting VLMs, such as SSL-cleanse[1] and DECREE[2].
Including a discussion on how BadVLMDriver could potentially evade these defenses would add depth to the paper’s security analysis.

[1] Zheng et al, SSL-cleanse: Trojan detection and mitigation in self-supervised learning. ECCV'24

[2] Feng et al., Detecting Backdoors in Pre-trained Encoders. CVPR'23

**Questions:**

Please respond to each weakness mentioned above.

---

> ### Author Response · Authors · 2024-11-21
> **Rebuttal (Part-I)**
>
> Thank you for reviewing our paper and for your encouraging comments. We especially appreciate your recognition of the importance of our work and the quality of our paper writing. We hope that our responses below will properly address your remaining concerns.
>
> ---
>
> **W1:** However, the novelty of the method may be limited, as it broadly follows the conventional backdoor attack paradigm by embedding malicious samples among clean data. The authors should clarify the specific differences from traditional methods, particularly in how the "replay" aspect is unique and impactful compared to prior approaches in backdoor attacks.
>
> **Response:** Thank you for your valuable advice. It is true that our weight poisoning backdoor attack shares similarities with conventional data poisoning backdoor attacks by mixing malicious samples with clean ones.
>
> However, our method is more effective and flexible because of the following two key differences:
>
> - **Correspondence between replayed and backdoor samples makes the tuning process more effective.** In traditional data poisoning attacks, backdoor samples lack clear correspondence with the clean ones. In comparison, our method tunes VLM on the generated backdoor training samples and their correspondent benign replays without the backdoor trigger and the backdoor target response. Such a correspondence amplifies the contrast between samples with and without the backdoor content, such that the backdoor mapping from the trigger to the target response will be easier learned. Our ablation study (Table 4 in the manuscript) demonstrates that without replay-based tuning, the VLM would generate the target behavior for almost all normal images that are without the trigger, making the backdoor attack highly detectable and thus unstealthy.
>
> - **Tuning solely on generated data enhances our attack's flexibility.** Traditional data poisoning attacks requires mixing backdoor samples into the original clean dataset, restricting the attack to pre-training stages (typically during the crowd-sourcing data annotation phase [1]). In contrast, our replay-based tuning leverages clean samples generated from the victim model, allowing the attack to rely entirely on generated data. This approach enables the attack to be carried out even after training is complete (e.g., during on-board local deployment), significantly increasing its flexibility.
>
> Beyond the "replay" aspect, BadVLMDriver introduces additional novelties, including **the use of daily physical objects as triggers and a language-guided automatic pipeline** to execute the attack. These differences from conventional backdoor attack paradigm significantly enhance the practicality, stealthiness, and efficiency of our method.
>
>
> [1] Manlu Shu et al. On the Exploitability of Instruction Tuning. Neurips, 2023.
>
> ---
>
> **W2:** The target VLMs evaluated are LLaVA and MiniGPT-4, which are not specifically tailored for autonomous driving applications. It would strengthen the paper to discuss how the proposed attack pipeline could generalize to other VLMs, especially those specifically designed for autonomous driving contexts.
>
> **Response:** We apologize for any misunderstanding caused and would like to kindly clarify that we our experiments in the manuscript have included CODA-VLM [1], a timely driving VLM specifically designed for autonomous driving applications. Furthermore, both LLaVA and MiniGPT-4, evaluated in our experiments, are fine-tuned on a well-recognized benchmark for driving VLMs [2] (we preserve their origin model configurations to evaluate BadVLMDriver's generalization ability across widely used VLM architectures). Table 1 in the manuscript demonstrate that our BadVLMDriver is effective across these VLMs.
>
>
> To further evaluate BadVLMDriver's performance on specialized driving VLMs, we conducted experiments on two additional autonomous driving VLMs from [2, 3] using real world images with two different types of physical triggers and target behaviors. The attack performance on these models, along with CODA-VLM from our manuscript, is presented in the following table. As shown in Table R1, **our attack pipeline continues to achieve a high success rate across these driving VLMs**, underscoring the robustness and versatility of our approach.
>
>  [**Table R1.** Real world evaluation on three autonomous driving VLMs. Our BadVLMDriver achieves high success rates on diverse driving VLMs across various triggers and target behaviors.]
> |  Trigger+Target    | Balloon+Accelerate | Balloon+Brake | Football+Accelerate | Football+Brake|
> | -------------     | ----- | -----  | ------- | ----- |
> | CODA-VLM | 92%  | 80%  | 88% | 92%  |
> | DriveLM | 81%  | 75%  | 84% | 84%  |
> | DriveLLaVA | 90%  | 80%  | 84% | 88%  |

---

> ### Author Response · Authors · 2024-11-21
> **Rebuttal (Part-II)**
>
> **W3:** The paper omits some recent backdoor defense strategies targeting VLMs, such as SSL-cleanse[1] and DECREE[2]. Including a discussion on how BadVLMDriver could potentially evade these defenses would add depth to the paper’s security analysis.
>
> **Response:** Thank you for your constructive suggestion. We acknowledge the relevance of backdoor defense strategies for pretrained vision encoder such as SSL-Cleanse [1] and DECREE [2], as VLMs directly apply these models trained by self-supervised learning to encode the input images. However, these two defenses are not effective against BadVLMDriver for the following reasons:
>
> - SSL-cleanse and DECREE are designed to **detect backdoors embedded in pretrained vision encoders, while BadVLMDriver does not change the parameters of these vision encoders.** These two methods assumes the vision encoder is attacked during pre-training stage using self-supervised learning techniques. However, BadVLMDriver only fine-tunes the vision-langauge connector and the language model, keeping the vision encoder clean. Thus, SSL-cleanse and DECREE would not be able to detect the backdoor embeded by BadVLMDriver.
>
> - SSL-Cleanse and DECREE are designed to **detect digital triggers, whereas BadVLMDriver employs physical objects as triggers.** The trigger inversion step in SSL-Cleanse and DECREE is limited to generating digital triggers that rely on pixel-wise modifications. Since BadVLMDriver uses physical objects as triggers, these trigger inversion methods cannot be directly applied to detect backdoors introduced by BadVLMDriver.
>
> [1] Mengxin Zheng et al. SSL-cleanse: Trojan detection and mitigation in self-supervised learning. ECCV, 2024.
>
> [2] Shiwei Feng et al. Detecting Backdoors in Pre-trained Encoders. CVPR, 2023.

---

> ### Author Response · Authors · 2024-11-22
> **Authors' Response**
>
> Dear Reviewer,
>
> We sincerely thank you for your quick response! We're pleased to hear that we've addressed some of your concerns. We would like to further address your remaining concerns as follows:
>
> ---
>
> To provide a clearer explanation, we mathematically illustrate the process of generating backdoor data $(I_{Backdoor}, R_{Backdoor})$ and replayed clean data $(I_{Clean} , R_{Replay})$ for attacking a clean victim model $\phi_{CleanVLM}$, with the selected backdoor trigger and target behavior in language $(L_{Trigger}, L_{Target})$.
>
> **Generation of Replayed Response from the Victim Model:** Given a clean image of a road scene without the backdoor trigger, $I_{Clean}$, the replayed response is generated using the clean victim model $\phi_{CleanVLM}$:
> $$R_{Replay} = \phi_{CleanVLM} (I_{Clean})$$
>
> Note that the clean image $I_{Clean}$ comes from an open-source road scene dataset independent from the original clean dataset used for training $\phi_{CleanVLM}$, since our weight-poisoning backdoor attack does not assume that the attacker has access to the original training dataset (which is the case in data-poisoning attacks).
>
> **One-to-One Correspondence between Backdoor and Replayed Samples:** For the same clean image $I_{Clean}$, we generate the corresponding backdoor sample $(I_{Backdoor}, R_{Backdoor})$ using a language-guided image editing model $\phi_{ImageEditing}$ to embed the trigger $L_{Trigger}$ into the image, and then applying a LLM $\phi_{LLM}$ to embed the target behavior $L_{Target}$ into the response:
>
> $$I_{Backdoor} = \phi_{ImageEditing} (I_{Origin}, L_{Trigger})$$
>
> $$R_{Backdoor} =  \phi_{LLM}(\phi_{CleanVLM} (I_{Backdoor}), L_{Target})$$
>
> This one-to-one correspondence ensures that the model not only learns the mapping from the backdoor triggers to target behaviors, but also keeps the mapping from clean samples to clean responses. Traditional data-poisoning backdoor attacks do not have such a correspondence, as they simply mix backdoor samples into the original clean dataset.
>
> These two key differences amplify BadVLMDriver's flexibility and effectiveness, making it applicable to a wider range of practical attack scenarios during the model supply chain compared with traditional data-poisoning attacks. This feature highlights the fact that simply keeping the original training dataset clean is not enough to ensure the safety of driving VLMs—the poisoning of model weights is also a significant source of risk.
>
> ---
>
> We greatly appreciate the reviewer for pointing out the unclear part in our writing. We will include these comparisons in the revised version of the paper.
>
> Thanks,
>
> Authors

---

> ### Author Response · Authors · 2024-11-23
> **Authors' Response**
>
> Dear Reviewer,
>
> Thank you for carefully reading our explanation and timely responding to us!
>
> ---
>
> Following your suggestion, we have added extra clarification of our threat model in **l.200-201 on page 4** of the revised manuscript. We appreciate your constructive suggestions that help us to improve our paper. Additionally, there is also a supplementary explanation in **Appendix B** on **page 16** of the original manuscript: "Data poisoning attack assumes that the attacker can only inject corrupted examples into the training set, typically during the crowd-sourcing annotation phrase. This assumption is reasonable for web applications like ChatGPT, since the service provider can keep the model on their private and trustworthy server. However, driving VLMs necessitate on-board, local deployment, exposing them to additional risks such as man-in-the-middle attacks. This context heightens the likelihood of weight poisoning attack. Therefore, our assumption that an attacker have the capability to access the model and alter part of its weight is reasonable in the driving scenario."
>
> In response to your question, we summerize the average attack success rate of three victim models in Table R1, as shown in Table R2 below. The results indicate that the ASR against DriveLM is slightly lower, suggesting that driving VLMs may exhibit greater robustness when their original training dataset includes data that is used for the attack. This observation underscores the effectiveness of our approach, as using a dataset independent of the original training set contributes to a higher attack success rate.
>
> [**Table R2.** The average attack success rate of three victim models in Table R1.]
> |                | CODA-VLM | DriveLM | DriveLLaVA |
> | -------------- | -------- | ----- | --------- |
> | Average Attack Success Rate|  88%        |   81%    |   85.5%        |
>
> ---
>
> Thank you once again for your time and insightful feedback!
>
> Sincerely,
>
> Authors

---

### Official Review · Reviewer_1UcC · 2024-11-03

**Soundness:** 3
**Presentation:** 3
**Contribution:** 3
**Rating:** 3
**Confidence:** 3

**Summary:**

This paper introduces BadVLMDriver, the first physical backdoor attack targeting vision large language models (VLMs) in autonomous driving. Using everyday objects as triggers, it induces unsafe driving decisions like sudden acceleration. Unlike pixel-level digital attacks, this method activates via real-world physical objects and is highly stealthy. Experiments on three VLM models, five triggers, and two behaviors show up to 92% success. The study underscores the threat to autonomous driving and the urgent need for robust defenses.

**Strengths:**

1. Utilizing common objects as triggers enhances the real-world feasibility of the attack.
2. The experiments cover various triggers, models, and behaviors, demonstrating broad applicability.
3. The study highlights potential security risks in current autonomous driving systems using VLMs.

**Weaknesses:**

1. Insufficient experimental diversity: More types of autonomous driving VLMs should be evaluated to fully understand the applicability and limitations of the method.
2. Lack of analysis on defense effectiveness: There is insufficient discussion and validation of how existing defense mechanisms respond to this attack.
3. Unverified effectiveness in complex driving environments: The effectiveness of the attack in complex or dynamic driving scenarios has not been adequately assessed.

**Questions:**

1. How does the presence of environmental factors (e.g., lighting, weather conditions) affect the attack's success rate?
2. Can the methodology be adapted to identify or mitigate other types of vulnerabilities in VLMs?

---

> ### Author Response · Authors · 2024-11-21
> **Rebuttal (Part-I)**
>
> We sincerely thank you for your time and efforts in reviewing our paper. We especially appreciate your recognition of the real-world feasibility, broad applicability and amplified social risk of our approach. We hope our responses below will alleviate your remaining concerns.
>
> ---
> **W1:** Insufficient experimental diversity: More types of autonomous driving VLMs should be evaluated to fully understand the applicability and limitations of the method.
>
> **Response:** Thank you for your valuable comment. We would like to kindly clarify that we have evaluated three distinct autonomous driving VLMs in the submission, each trained on representative driving-related datasets [1, 2]. These VLMs cover three typical VLM architectures and training pipelines, ensuring a comprehensive assessment of our method. As shown in Table 1 of the manuscript, the results demonstrate that our attack generalizes effectively across driving VLMs with diverse structures and training pipelines.
>
> Additionally, following your suggestions, we further strengthen the experimental diversity by evaluating two additional autonomous driving VLMs from [1, 3] using real world images with two different types of physical triggers and target behaviors. The attack performance on these models, along with CODA-VLM from our manuscript, is presented in the following table. As shown in Table R1, **our attack pipeline continues to achieve a high success rate across these driving VLMs**, underscoring the robustness and versatility of our approach.
>
>  [**Table R1.** Real world evaluation on three autonomous driving VLMs. Our BadVLMDriver achieves high success rates on diverse driving VLMs across various triggers and target behaviors.]
> |  Trigger+Target    | Balloon+Accelerate | Balloon+Brake | Football+Accelerate | Football+Brake|
> | -------------     | ----- | -----  | ------- | ----- |
> | CODA-VLM | 92%  | 80%  | 88% | 92%  |
> | DriveLM | 81%  | 75%  | 84% | 84%  |
> | DriveLLaVA | 90%  | 80%  | 84% | 88%  |
>
> [1] Chonghao Sima et al. DriveLM: Driving with Graph Visual Question Answering. ECCV, 2024.
>
> [2] Kai Chen et al. Automated Evaluation of Large Vision-Language Models on Self-driving Corner Cases. WACV, 2025.
>
> [3] Rui Zhao et al. DriveLLaVA: Human-Level Behavior Decisions via Vision Language Model. Sensors, 2024.

---

> ### Author Response · Authors · 2024-11-21
> **Rebuttal (Part-II)**
>
> **W2:** Insufficient discussion and validation on existing defenses.
>
> **Response:** Thank you for your valuable comment. We have provided a discussion and validation on the resilience of BadVLMDriver against existing defense mechanisms in Section 4.4 and Appendix C of the manuscript, covering rule-based filtering, noise reduction mechanisms, existing backdoor defenses and incremental finetuning.
>
> Following your suggestions, we have expanded on these discussions and conducted additional experiments to provide a more comprehensive discussion and validation.
>
> - **Rule-based filtering** is ineffective since our BadVLMDriver allows for flexible selection of both the backdoor trigger and the malicious target behavior, making it challenging for rule-based systems to account for all possible attack scenarios. For example, recent LLM-based driving system [1] perform collision checks with pedestrians and vehicles, yet they fail to prevent attacks that induce sudden braking, which could cause rear-end collisions, or sudden acceleration upon encountering a football, posing a risk of harm to unseen children in blind spot chasing the ball.
>
> - **Noise reduction mechanisms** [2] also fall short, as they are designed to mitigate perturbations used in digital attacks. BadVLMDriver employs physical objects as triggers, which are not mitigated by noise reduction mechanisms typically designed to counteract perturbation patterns added to images. We tested the attack using mean filtering and median filtering algorithms on real-world images, with results in Table R2 demonstrating that these methods do not reduce the attack success rate.
>
> - **Existing backdoor defense strategies** are not applicable to VLMs. Most of the current work in this area targets image or language classifiers [3], which assume a finite and discrete output space (e.g., image or sentiment classification). While recent backdoor detection methods for pre-trained image encoders [4] do not rely on this assumption, they still cannot effectively defend against our attack, as they are designed to detect backdoors embedded in the vision encoder’s weights, which remain unchanged during our attack. Although there is a recent defense specifically targeting weight poisoning backdoor attacks [5], its application is limited to LLMs. As shown in Table R2, this defence can not reduce the success rate of our attack.
>
> - **Anti-jailbreak defenses** [7] introduce unacceptable latency for driving systems by employing additional clean models for verification and multi-round discussions. For instance, ECSO [7] requires a four-round self-reflection process during inference, resulting in significant extra latency. While System Prompt-based Defense (SPD) [6] avoids excessive computational overhead by only adding an additional prompt, the results in Table R2 indicate that SPD fails to defend against our attack.
>
> - **Incremental fine-tuning** on clean datasets can reduce the attack success rate by forcing the model to catastrophically forget the backdoors hidden in the parameters, as shown in Appendix C. However, this approach is only a partial solution. It remains ineffective when the attacker carries out post-training attacks, such as manipulating the model's weights during the local on-board deployment stage.
>
> From the results and discussion above, we see that 1) our physical backdoor attack BadVLMDriver is robust against noise reduction mechanisms and existing backdoor defense strategies; 2) anti-jailbreak defenses introduce significant computational overhead, making them impractical for real-time decision-making systems in autonomous vehicles; 3) rule-based filtering and incremental fine-tuning are only partial solutions due to the stealthiness and flexibility of our BadVLMDriver, as they can not cover the wide range of triggers, target behaviors and attack scenarios supported by our language-guided automatic attack pipeline.
>
> [**Table R2.** Current noise reduction, backdoor defense and anti-jailbreak defenses can not reduce the attack performance of BadVLMDriver. ]
> | | No defense|Mean filtering|Median filtering|PSIM [5]|SPD [6]|
> |-|-|-|-|-|-|
> | Attack success rate ↑ |92%|92%|92%|92%|92%|
>
> [1] Jiageng Mao et al. A Language Agent for Autonomous Driving. COLM, 2024.
>
> [2] Erwin Quiring et al. Adversarial preprocessing: Understanding and preventing image-scaling attacks in machine learning. USENIX Security Symposim, 2020.
>
> [3] Kunzhe Huang et al. Backdoor Defense via Decoupling the Training Process. ICLR, 2022.
>
> [4] Shiwei Feng et al. Detecting Backdoors in Pre-trained Encoders. CVPR, 2023.
>
> [5] Shuai Zhao et al. Defending Against Weight-Poisoning Backdoor Attacks for Parameter-Efficient Fine-Tuning. Arxiv, 2024.
>
> [6] Yunhao Gou et al. Eyes Closed, Safety On: Protecting Multimodal LLMs via Image-to-Text Transformation. ACL, 2024.
>
> [7] Siyuan Ma et al. Visual-RolePlay: Universal Jailbreak Attack on MultiModal Large Language Models via Role-playing Image Character. Arxiv, 2024.

---

> ### Author Response · Authors · 2024-11-21
> **Rebuttal (Part-III)**
>
> **W3:** Unverified effectiveness in complex or dynamic driving scenarios.
>
> **Response:** Thank you for your valuable comment. We would like to kindly clarify that the two datasets used in our experiments exactly represents complex and dynamic driving scenarios.
>
>  - The benchmark dataset, nuScences [1], contains a diverse range of **urban, suburban, and highway environments, each with various weather conditions, lighting changes, and traffic densities**. The dataset includes over 1,000 scenes recorded from a full 360-degree sensor suite, which provides detailed data on both **static and moving objects**, such as vehicles, pedestrians, and cyclists. This diversity enables comprehensive testing across different driving situations, from dense city traffic to open highways.
>
>  - Our self-collected dataset also effectively captures the complexity and dynamics of real-world driving scenarios, as shown in Figures 4, 7, 8, and 9 of the manuscript. The dataset accounts for **dynamic traffic participants**, such as pedestrians with balloons, vehicles passing traffic cones, motorcycles near soccer balls, and children playing with footballs. It also includes **triggers at varying distances and angles from the camera**, reflecting the dynamic interactions between the autonomous vehicle and its environment. Additionally, in response to the reviewer’s first question, we evaluated the data under **different lighting and weather conditions**, covering scenarios such as clear/rainy days and nights (both near and away from streetlights), ensuring a comprehensive representation of real-world driving challenges.
>
> BadVLMDriver demonstrates a high attack success rate on these datasets, as shown in Table 1&2 of the manuscript and the following response to Q1, highlighting its effectiveness in complex and dynamic driving environments.
>
> [1] Holger Caesar et al. nuScenes: A Multimodal Dataset for Autonomous Driving. CVPR, 2020.
>
> ---
> **Q1:** How does the presence of environmental factors (e.g., lighting, weather conditions) affect the attack's success rate?
>
> **Response:** Thanks for raising this valid concern. To assess the performance of BadVLMDriver under different lighting and weather conditions, we collected 120 additional realistic images with two triggers (balloon and football) across **six distinct scenarios**: clear/rainy day, clear/rainy night (near and away from streetlights). Sample images for each scenario can be found in **Figure 12-13 on page 22-23** of the revised manuscript. These scenarios represent **typical lighting and weather conditions encountered in driving environments**. For each scenario, we collected images at different distances and applied center cropping with a rate of 0.7 and 0.9 to augment the dataset. Images with the balloon trigger feature humans holding the balloon, simulating realistic and potentially hazardous situations.
>
> Results in Table R3 below demonstrate that: 1) BadVLMDriver maintains a high attack success rate across different weather and lighting conditions. 2) In rainy weather or under poor lighting conditions (e.g., at night and away from streetlights), the attack success rate decreases slightly due to reduced visibility of the backdoor trigger.
>
>  [**Table R3.** Attack success rate in different lighting and weather conditions. BadVLMDriver continues to achieves a high attack success rate in various conditions.]
> |  Weather | Lighting   | Football+Accelerate | Football+Brake | Balloon+Accelerate | Balloon+Brake |
> |-|-|-|-|-|-|
> | Clear | Day | 100%  | 100%  | 87% | 100%  |
> | Clear |Night / Near Light | 100%  | 100%  | 77% | 93%  |
> | Clear |Night / Away from Light | 87%  | 90%  | 77% | 80%  |
> | Rainy | Day | 100%  | 100%  | 87% | 90%  |
> | Rainy |Night / Near Light | 90%  | 80%  | 77% | 80%  |
> | Rainy |Night / Away from Light | 70%  | 80%  | 80% |73%|
>
> ---
> **Q2:** Can the methodology be adapted to identify or mitigate other types of vulnerabilities in VLMs?
>
> **Response:** Yes, BadVLMDriver can be extended to implement clean-label backdoor attack against VLMs in other safety-critical applications, such as those used for robot manipulation [1] or moderating images [2] with toxic content. These attacks can evade existing defense mechanisms due to their stealthiness and flexibility, leading to severe consequences, such as inducing harmful actions in robotic systems or allowing violent and gory content to bypass moderation filters and be uploaded to social media.
>
> This highlights the need for VLM producers to address the risks of such harmful physical backdoor attacks. Specifically, attention should be given to attacks involving everyday objects as triggers, and safeguards should be in place to ensure that model weights are not tampered with by attackers.
>
> [1] Beichen Wang et al. VLM See, Robot Do: Human Demo Video to Robot Action Plan via Vision Language Model. Arxiv, 2024.
>
> [2] Mamadou Keita et al. Harnessing the Power of Large Vision Language Models for Synthetic Image Detection.  ICASSP, 2024.

---

> > ### Author Response · Authors · 2024-12-01
> > **We sincerely anticipate your feedback as the Discussion stage will end in 3 Days.**
> >
> > Dear Reviewer 1UcC,
> >
> > Thank you again for your valuable feedback.
> >
> > We have addressed your suggestions by incorporating additional experiments and discussions. As the Discussion Stage will conclude in 3 days, we would greatly appreciate it if you could review our responses and let us know if there are any remaining points of clarification.
> >
> > Your recognition is really vital to us.
> >
> > Best regards,
> >
> > Authors

---

> ### Author Response · Authors · 2024-11-25
> **Your recognition is vital to us**
>
> Dear Reviewer,
>
> Thank you once again for the time you dedicated to reviewing our paper and for the invaluable feedback you provided!
>
> We hope our responses have adequately addressed your previous concerns. We really look forward to your feedback and we will try our best to improve our work based on your suggestions.
>
> Sincerely,
>
> Authors

---

> ### Author Response · Authors · 2024-12-03
> **Your feedback is invaluable to us.**
>
> Dear Reviewer 1UcC,
>
> We greatly appreciate your feedback on our work. In response to your suggestions, we have expanded our experiments to include a wider range of driving VLMs and environmental factors, provided more in-depth analysis on the effectiveness of existing defense methods, and clarified the dataset used in our manuscript.
>
> As the Discussion Stage will **end in 4 hours**, we kindly request that you review our updated responses and reconsider your rating.
>
> Thank you for your time and consideration.
>
> Best regards,
>
> Authors

---

### Meta-Review · Area_Chair_xvKS · 2024-12-21

**Metareview:**

This paper proposes BadVLMDriver, a backdoor attack method against VLMs for autonomous driving. To enhance practicality, the authors use common physical objects (a red balloon), to initiate unsafe actions like sudden acceleration, highlighting a real-world threat to autonomous vehicle safety. The authors validate their approach through extensive experiments across various triggers, achieving a high 92% attack success rate with a low false attack rate. The reviewers mentioned the following strengths: (1) The paper addresses a highly relevant topic, focusing on safety risks in autonomous driving posed by Vision-Large-Language Models (VLMs). (2) It leverages real-world physical objects—such as a red balloon—to trigger malicious behaviors in autonomous vehicles. (3)  The paper is well written. However, , reviewers also mentioned some key limitations: (1) Limited novelty. The work did not show a significant difference to existing backdoor attacking methods like BadNets, which is like applying existing works to the new setups. (2) Narrow tasks. The work only focuses on the VQA task instead of autonomous driving-related tasks and existing backdoor attacks on general VQA models could potentially achieve the same effect.

After reviewing the paper and the reviewers' comments, I agree that the concerns raised are both reasonable and significant for a top-tier publication. However, I believe the following points also warrant attention: (1) The motivation for backdoor attacks on autonomous driving visual-language models (VLMs) is not sufficiently compelling. In practice, autonomous driving systems are unlikely to rely on open-source models that could be vulnerable to backdoor attacks. (2) Existing research has already demonstrated that diffusion-based generation methods can create physical adversarial patches against pre-trained VLMs. This approach is not only more practical but also more relevant than employing backdoor attacks on autonomous driving VLMs. We encourage the authors to improve the paper to address the concerns.

**Additional Comments On Reviewer Discussion:**

All reviewers provided thoughtful and constructive feedback in the first round of review. Two reviewers gave clear acceptance scores and acknowledged the significance of the work, but their confidence scores were 3. This suggests that they may have had difficulty understanding certain aspects of the submission or were unfamiliar with some related work. Reviewer 1UcC noted that the submission focuses on narrow tasks, which detracts from its novelty. Although Reviewer 1UcC also gave a confidence score of 3, their comments were supported by Reviewer 9XEN. Given the variance in scores, I have carefully reviewed the paper and the comments. The concerns raised should be thoroughly addressed in the revised submission.

---

### Decision · Program_Chairs · 2025-01-22

Reject